# Study on the Micro Removal Process of Inner Surface of Cobalt Chromium Alloy Cardiovascular Stent Tubes

**DOI:** 10.3390/mi13091374

**Published:** 2022-08-23

**Authors:** Zhuang Song, Yugang Zhao, Zhihao Li, Chen Cao, Guangxin Liu, Qian Liu, Xiajunyu Zhang, Di Dai, Zhilong Zheng, Chuang Zhao, Hanlin Yu

**Affiliations:** Institute for Advanced Manufacturing, Shandong University of Technology, Zibo 255049, China

**Keywords:** magnetic abrasive finishing, magnetic abrasive powder, cobalt–chromium alloy cardiovascular stent tube, surface roughness, material removal thickness

## Abstract

Due to the special manufacturing process of cobalt–chromium alloy cardiovascular stent tubes, there are serious surface defects in their inner walls, which affects the therapeutic effect after implantation. At the same time, the traditional processing technology cannot finish the inner wall of a cardiovascular stent tube. In light of the above problems, magnetic abrasive finishing (MAF) equipment for the inner wall of an ultra-fine and ultra-long cardiovascular stent tube is proposed, and MAF technology is used to improve the surface quality of its inner wall. High-performance spherical magnetic abrasive powders are used to finish the inner wall of a cobalt–chromium alloy cardiovascular stent tube with an inner diameter of 1.6 mm and an outer diameter of 1.8 mm. The effects of finishing time, tube rotational speed, feed speed of the magnetic pole, MAPs filling quantity, and MAP abrasive size on the surface roughness and material removal thickness of cobalt–chromium alloy cardiovascular stent tube are investigated. The results show that the surface roughness of the inner wall of the cobalt–chromium alloy cardiovascular stent decreases from 0.485 μm to 0.101 μm, and the material removal thickness of the defect layer is 4.3 μm. MAF technology is used to solve the problem of the poor surface quality of the inner walls of ultra-fine and ultra-long cobalt–chromium alloy cardiovascular stent tubes.

## 1. Introduction

Nowadays, the high incidence of cardiovascular diseases seriously endangers human life and health [1,2,3]. The methods for the treatment of cardiovascular diseases mainly include drug treatment, surgical treatment, and interventional treatment. Interventional treatment is the implanting of dilated cardiovascular stents in the lesion site by medical means, so that the blood vessels can restore the smooth state of blood delivery [4,5,6]. This treatment method has the advantages of a good therapeutic effect and a low treatment cost; it is used by the majority of patients.

Cobalt–chromium alloy has good mechanical properties, biocompatibility, and corrosion resistance and has been playing an important role in the medical field [7,8,9,10]. At the same time, cobalt–chromium alloy stents have good *X*-ray visibility, so they are widely used in the production of cardiovascular stents. Cobalt–chromium alloy cardiovascular stent tubes have the characteristics of slenderness and its production process including rolling, extrusion, drawing, heat treatment, etc., so its inner wall can not avoid the existence of wrinkles, cracks, scratches, pits, and other surface defects or even the appear of a metamorphic layer of hard oxide particles resulting in the high surface roughness of its inner wall. After these cardiovascular stents with metamorphic layers and surface defects are implanted into the human body, it is easy to cause the secondary formation of thrombosis under the long-term flow of blood. Therefore, improving the surface quality of the inner wall of the tube has become an urgent problem to be solved.

Because of the small inner diameter and the long length of a cobalt–chromium alloy cardiovascular stent tube, a traditional finishing tool cannot penetrate the tube. At the same time, the outer wall can not be damaged in the finishing process, which leads to the traditional processing method of not removing the defective layer of the inner wall. At present, most researchers use an electrochemical polishing method to finish cardiovascular stent tubes [11,12]. However, it is easy to produce defects such as pitting, bulges, and an oxide layer on the surface after finishing and integrating toxic elements, which causes new harm to the human body.

Magnetic abrasive finishing (MAF) technology is a very effective ultra-precision machining method, which is widely used in finishing difficult-to-machine materials and complex surfaces. Hanada et al. [13] prepared spherical metal-based diamond magnetic abrasive powders with a plasma-spraying technique to solve the fine processing of the inner surface of the capillary. At the same time, SUS304 steel plate is processed by MAF to obtain the surface roughness *R_t_* of 77.4 nm. Naveen et al. [14] developed a four-pole magnetic tool and placed an auxiliary pole under the workpiece, which increased the magnetic-induction intensity to 1.8 times the previous one. The optimized process parameters were used for 24 passes of MAF of SS304 material, and the surface roughness was reduced from 0.3 μm to 0.022 μm. Singh et al. [15] found that using a mechanical alloying technique could prepare more uniform MAPs at lower temperature. At the same time, it was found that the annealing temperature could affect the performance of the MAP. The MAP annealed at 950 °C had excellent magnetic conductivity, and the MAP annealed at 1050 °C had excellent finishing performance. After 60 min of MAF of the brass tube internal surface with this MAP, the average surface roughness was reduced to 0.51 μm. Heng et al. [16] used MAF technology to precisely manufacture ZrO_2_ ceramic rods with medium-sized diameters. According to the finite element simulation and experiment, a 2 mm square magnetic pole is more suitable for surface processing. Finally, after 40 s MAF, the surface roughness *Ra* of ZrO_2_ ceramic rods decreases from 0.18 μm to 0.02 μm and the roundness error decreases from 3.5 μm to 0.2 μm. Fan et al. [17] polished the difficult-to-machine Ti-6Al-4V workpiece using MAF technology. The effects of intelligent shear-thickening-fluids concentration, working gap, feed rate, and spindle rotational speed on its surface quality are explored through experiments. Finally, the surface roughness, *Ra,* decreases from from 1.17 μm to 54 nm, and the nano-machining of difficult-to-machine materials was realized. Sun et al. [18] proposed a new type of magnetic pole that can generate alternating magnetic field in the processing area and explored the feasibility of the new magnetic pole through a MAF experiment and finite element simulation on the surface of 316 L stainless steel printed by selective laser melting (SLM). The surface roughness, *Ra,* of 316 L stainless steel printed by SLM decreased from 9.79 μm to 0.85 μm. In order to improve the wear resistance and fatigue performance of cobalt–chromium alloy cardiovascular stent tubes, MAF technology is used to finish the inner walls of the tubes to remove the surface defect layers of the inner walls and improve the surface quality, as well as to provide favorable conditions for subsequent coating.

In this study, MAF is used to remove the defect layer of the inner wall and improve its surface quality. Firstly, the principle and material-removal mechanism of MAF for inner wall of cardiovascular stent are introduced. Secondly, according to the MAF principle, the MAF setup for the inner wall of ultra-fine and ultra-long cardiovascular stent tubes is proposed. Cubic boron nitride (CBN) hard abrasive grains are more suitable for machining metal materials, so the CBN/metal spherical magnetic abrasive powders with far superior performance than the traditional preparation method are prepared by a gas–solid two-phase double-stage atomization and rapid solidification method. Then, a single-factor experiment is designed to explore the influence of process parameters on the surface roughness *Ra* and removed-material thickness of the inner wall. Finally, the defect layers on the inner walls of cobalt–chromium alloy cardiovascular stent tubes are removed, and surface quality is significantly improved.

## 2. Mechanism of Magnetic Finishing for Tube Inner Wall

### 2.1. Principle of Processing

Figure 1 shows the processing principle of MAF on the inner wall of a cobalt–chromium alloy cardiovascular stent tube. The appropriate amount of MAP is filled into the tube, and the gap on both sides of the tube is sealed and clamped on the precision fixture. The MAP in the tube is magnetized instantaneously into a ‘flexible magnetic abrasive brush (FMAB)’ with micro cutting ability under the action of an external magnetic field. The FMAB also has certain flexibility, which can imitate the complex shape of the inner wall of the tube and attach to its surface. The cardiovascular stent tube rotates under the drive of the motor, and the FMAB moves relative to the inner wall of the tube, resulting in friction, cutting, plowing, and other effects on the inner wall of the tube. The removal of the defect layer on the inner wall of the tube is realized, and the surface roughness is finally decreased.

MAPs are arranged along the direction of the magnetic force line, and each MAP is a cutting tool with cutting ability. The force analysis of a single MAP in Figure 1 shows that the MAP is subjected to *F_x_* in the direction of the magnetic force line and *F_y_* in the direction of the equipotential line under the action of the magnetic field, and their resultant force *F_m_* provides the pressure required in the process of MAF [19,20]. The calculation formula is shown in Equation (1).
(1)$$\left\{\begin{array}{l} F_{x}=kV_{0}x_{m}H\frac{\partial H}{\partial x}\\ F_{y}=kV_{0}x_{m}H\frac{\partial H}{\partial y}\\ F_{m}=\sqrt{F_{x}{}^{2}+F_{y}{}^{2}} \end{array}\right.$$

The magnetic induction intensity *B* is proportional to the magnetic field intensity, *H*, and the relationship is shown in Equation (2).
(2)$$H=\frac{B}{\mu_{0}\cdot (1+x_{m})}$$
where *k* is the magnetization coefficient, *V*_0_ is the abrasive volume, *x_m_* is the magnetic susceptibility of MAP, *μ*_0_ is vacuum permeability, and $\frac{\partial H}{\partial x}$ and $\frac{\partial H}{\partial y}$ are the gradient of the magnetic field intensity along the magnetic force line and the equipotential line, respectively. Based on the above equations, the magnetic field force *F_m_* is shown in Equation (3).
(3)$$F_{m}=\frac{\pi kd_{m}{}^{3}Bx_{m}}{6\mu_{0}(1+x_{m})}\sqrt{{\left(\frac{\partial H}{\partial x}\right)}^{2}+{\left(\frac{\partial H}{\partial y}\right)}^{2}}$$
where *d_m_* is the particle size of MAP.

The magnetic-induction intensity is an important parameter in the MAF process [21,22,23]. Magnetic field force generation under the MAF action provides the pressure required in the MAF process and forms the normal pressure to remove the material on the inner wall of the tube. According to Equation (3), magnetic field force is proportional to magnetic-induction intensity. When the magnetic-induction intensity increases, the MAP can produce enough normal pressure to better press into the inner wall defect layer. At the same time, the MAP can flow regularly with the magnetic field gradient, which makes each MAP able to participate in the MAF process. The FMAB will also be better bound in the processing area, so that it will not appear in a chaotic state with a high tube rotational speed. Finally, effective finishing processing is realized to achieve a better material-removal effect.

### 2.2. Material-Removal Mechanism

The removal mechanism of cardiovascular stent tube by MAF is the micro-cutting and plastic deformation of the inner wall surface by hard abrasive particles embedded in the metal substrate surface [24]. A schematic of material removal is shown in Figure 2. Under the action of the magnetic field, there is a vertical downward normal pressure, *F_n_*, for the MAP pressed on the inner wall surface of the tube, which makes the hard abrasive produce a certain indentation depth on the inner wall surface [25]. At the same time, the metal matrix in the MAP will also have a good extrusion effect on the surface of the workpiece, which can change the stress state of the tube surface. At this point, the single hard abrasive embedded on the surface of each MAP will bear the normal pressure, *F_n_*, and the tangential cutting force, *F_t_*, imposed by the MAP. The hardness of hard abrasive particles is generally higher than that of processed materials. Under the action of normal pressure and tangential cutting force, the blade tip of abrasive particles will have a cutting effect on the defect layer. In this process, the tube inner wall is removed layer by layer from the sharp convex point. It can be seen that the amount of material removed chanes from more to less and ultimately becomes stable.

## 3. Setup and Materials

### 3.1. MAF Setup

Figure 3 is the MAF setup for the inner wall of an ultra-fine long cardiovascular stent tube. The setup is mainly composed of a clamping system, magnetic field generating system, transmission system, and control system. The servo motor in the clamping system can drive the tube to rotate. The magnetic-field-generation system is equipped with slotted permanent magnet poles. The slotted permanent magnet poles can release a more uniform magnetic field, making the inner wall of the tube more uniform. The transmission system can adjust the position of the magnetic-field-generating system with a stepping motor. The control system adjusts the speed and direction of the tube by controlling the servo motor, so as to adjust the magnetic abrasive trajectory, which is more suitable for the processing of cobalt chromium alloy cardiovascular stent. The feed speed and direction of the magnetic-field-generation system are adjusted by controlling the stepper motor. Each system works coordinately to realize the MAF of the inner wall of the tube.

### 3.2. Experimental Magnetic Abrasive Powders

Magnetic abrasive powder plays a key role in the MAF process. It is a metal-based particle-reinforced composite powder from a composition point of view. The metal-based phase exhibits magnetic conductivity, and the hard abrasive phase exhibits grinding performance. CBN has the advantages of high hardness, high compressive strength, and high thermal conductivity, which makes the machined surface difficult to crack and burn. In addition, the chemical inertness of CBN is strong, and it is not easy to react with iron-group elements. Therefore, for the finishing of the cobalt–chromium alloy, the selection of CBN hard abrasive particles can obtain a better surface quality.

In order to solve the problem of the poor performance of MAPs prepared by traditional methods, CBN/metal MAPs with different particle sizes are prepared by a gas–solid two-phase double-stage atomization and rapid solidification method. This new method can prepare high-performance MAPs with high efficiency and low cost. The molten metal flow flows into the atomization chamber under the action of gravity. In this process, the low-pressure inert gas flow containing hard abrasive particles is first encountered, and the hard abrasive particles break through the molten metal surface. It then encounters high-pressure inert gas flow and is atomized into tiny droplets containing hard abrasive particles [26]. The SEM images of the prepared CBN/metal MAP are shown in Figure 4. It can be seen that the CBN hard abrasive is uniformly and densely embedded on the surface of the iron matrix and that the CBN/metal MAP is ideally spherical. Spherical MAP has special properties such as a micro-edge. In the MAF process, the abrasive particles keep the same cutting depth on the surface of the tube, which greatly improves the efficiency of MAF [27].

### 3.3. Experimental Cobalt–Chromium Alloy Cardiovascular Stent Tube

A physical diagram of a cobalt–chromium alloy cardiovascular stent tube used in the experiment is shown in Figure 5, with an outer diameter of 1.8 mm and an inner diameter of 1.6 mm. The composition and properties of cobalt–chromium alloy cardiovascular stent tubes are shown in Table 1 and Table 2. The length of the tube used in the experiment was long. In order to save the experimental cost of the tube, the tube was cut into 500 mm short tubes for each group of experiments.

## 4. Results and Discussion

### 4.1. Experimental Details

In order to explore the influence of the process parameters of MAF technology on the surface quality of a cobalt–chromium alloy cardiovascular stent tube, the effects of finishing time, tube rotational speed, feed speed of magnetic pole, MAPs filling quantity and MAP particle size on the material-removal thickness and surface roughness *Ra* of the inner wall of cobalt–chromium alloy cardiovascular stents are studied with a single-factor experiment. Details of the selection of each parameter level are shown in Table 3, and its scope is based on experimental setup, experimental materials, and previous research progress.

In order to reduce the error of the experiment and improve the reliability of the experiment, first of all, a section of tube was intercepted and cut. Five measurement-points were evenly dispersed on the inner wall of the tube in each group of experiments. The surface roughness *Ra* of the five points was measured by a 3D digital microscope (DSX1000, OLYMPUS, Tokyo, Japan), and the average value was taken as the initial surface roughness of the inner wall of the tube. The calculated value is 0.485 μm. Then the CBN/metal MAPs were injected into the tube, and the oil-based grinding fluid was injected as the processing medium. The magnetic pole position and control system were debugged, and the two ends of the tube were sealed and clamped on the servo motor to start the experiment. After the end of the experiment, the tube was removed, and the impurities in the inner wall of the tube were cleaned by anhydrous ethanol through an ultrasonic cleaning instrument. The quality of the tube was measured after drying. Finally, the same method was used to measure the surface roughness *Ra* of the inner wall of the tube after MAF.

The thickness of the defect layer on the inner wall of a cobalt–chromium alloy cardiovascular stent is about 4μm. Therefore, in addition to the surface roughness, the material-removal thickness of the inner wall of the tube is also a very important characterization index that can determine whether the MAF can effectively remove the deterioration and defect surface of the inner wall of the tube. The material-removal thickness of the inner wall of the tube can be determined by the material-removal amount of the tube before and after MAF. The formula is shown in Equation (4).
(4)$$\left\{\begin{array}{l} M_{1}=\frac{\left(\pi d_{1}^{2}-\pi d_{2}^{2}\right)\rho L}{4}\\ M_{2}=\frac{\left(\pi d_{1}^{2}-\pi d_{3}^{2}\right)\rho L}{4}\\ \Delta M=M_{1}-M_{2} \end{array}\right.$$
where *M*_1_ is the weight of tube before MAF and *M*_2_ is the weight of the tube after MAF; Δ*M* is the material removal amount of tube before and after MAF; *d*_1_ is the outer diameter of the tube; *d*_2_ is the inner diameter of the tube; *d*_3_ is the inner diameter of the tube after MAF; *ρ* is the density of tube; *L* is the length of the experimental tube. According to Equation (4), the inner diameter, *d*_3_, of the tube after MAF is shown as Equation (5).
(5)$$d_{3}=\sqrt{\frac{4\cdot \Delta M}{\rho \pi L}+d_{2}^{2}}$$

Thus, the calculation formula of the material-removal thickness, Δd, of the inner wall of the cardiovascular stent tube is shown in Equation (6).
(6)$$\Delta d=d_{2}-d_{3}=d_{2}-\sqrt{\frac{4\cdot \Delta M}{\rho \pi L}+d_{2}^{2}}$$

### 4.2. Effect of Finishing Time

According to the principle of the single-factor experiment, a different finishing time is selected, and the tube rotational speed, the feed speed of magnetic pole, the MAP abrasive size, and the MAPs filling quantity are fixed. The effect of finishing time on the surface roughness and material removal thickness of the inner wall of the cobalt–chromium alloy cardiovascular stent tube is explored. The experimental results are shown in Figure 6.

The specific process parameters were as follows: the tube rotational speed was 700 r/min; the feed speed of the magnetic pole was 150 mm/min; the MAPs filling quantity was 0.2 g; the MAP particle size was 150μm, and the finishing time was 120–600 min. According to the experimental results, it is obvious that material-removal thickness increased with the increase in finishing time, and the CBN/metal MAPs used in this experiment still maintained its micro cutting ability after lengthy MAF. After 360 min of MAF, the inner wall material-removal thickness was more than 4 μm, indicating that the defect layer had been basically removed. The surface roughness, with the increase in finishing time, showed a trend of first decreasing and then increasing. When the finishing time was 360 min, the surface roughness reached 0.108 μm, which is the lowest value. This may be because the finishing time was too long and the inner wall was over-processed after removing the defect layer, so that the material is excessive removed and the inner wall is damaged again, which eventually leads to the increase in the surface roughness again.

### 4.3. Effect of Tube Rotational Speed

At different tube rotational speeds, other process parameters are fixed at the intermediate level, and the experimental results after 360 min are shown in Figure 7. Experiments show that, when the tube rotational speed is 700 r/min–900 r/min, the surface roughness *Ra* and the material-removal thickness are ideal. Under the existing process parameters, when the tube rotational speed is 900 r/min, the surface roughness reaches a minimum of 0.101 μm. When the tube rotational speed is too small, the number of interactions between the hard abrasive particles and the inner wall of the tube is reduced, which makes the defect layer unable to be completely removed after the same duration of MAF. When the tube rotational speed exceeds 900 r/min, the material-removal thickness decreases rapidly. This shows that, when the rotational speed is too high, the cutting force generated by the hard abrasive particles on the inner wall of the tube will be less than the friction force, which makes the FMAB appear in a chaotic state and means that the defect layer on the inner wall of the tube cannot be fully removed. It has been found through experiments that the tube rotational speed has a great influence on the surface quality of the inner wall of the tube, and a moderate tube rotational speed should be selected in the processing.

### 4.4. Effect of Feed Speed of Magnetic Pole

At different feed speeds of magnetic pole, other process parameters are fixed at the intermediate level, and the experimental results after 360 min are shown in Figure 8. According to the existing experimental results, the feed speed of the magnetic pole is the most appropriate from 100 mm/min to 150 mm/min, and the surface roughness *Ra* is the lowest at 100 mm/min. With the acceleration of the feed speed of the magnetic pole, the material-removal thickness decreases and the surface roughness increases. This is because, when the feed speed of the magnetic pole is too fast, the MAP at the end of the FMAB will lag, resulting in the number of MAPs involved in MAF being less and less and in the material-removal efficiency being reduced. At the same time, the MAP that is out of the magnetic field bound due to the hysteresis phenomenon appears in a chaotic state under the high tube rotational speed, which is easy to make the finished surface scratched again, resulting in the surface roughness rising again. The feed speed of the magnetic pole should not be too slow, because too slow of a feed speed of the magnetic pole will increase the path length of repeated machining, which causes the MAP to induce excessive material removal in the inner wall of the tube.

### 4.5. Effect of Abrasive Size

Different magnetic abrasive sizes are selected, and other process parameters are fixed at the intermediate level. The experimental results after 360 min are shown in Figure 9. According to the experimental results, too large or too small of an abrasive size will not obtain low surface roughness; when the abrasive size is 150 μm, the surface roughness reaches a minimum of 0.108 μm. The material-removal thickness increases with the increasing abrasive size. According to Equation (3), under the same experimental conditions, the larger the abrasive size is, the greater the normal pressure on the inner wall of the tube and the greater the indentation depth. In the initial stage of MAF, a larger abrasive size can more efficiently remove the defect layer on the inner wall of the tube. However, when the defect layer is removed, the larger normal pressure causes secondary scratches on the inner wall of the tube again. At the same time, the spiral trajectory generated by the larger abrasive size in the machining process is relatively loose, which is not conducive to more precise machining. However, too small of an abrasive size cannot produce an appropriate indentation depth, resulting in the incomplete removal of the defect layer. Therefore, different abrasive sizes should be selected according to the hardness of the material.

### 4.6. Effect of Magnetic Abrasive Filling Quantity

Weighing different magnetic abrasive filling quantities and fixing other process parameters at the intermediate level, the experimental results after 360 min are shown in Figure 10. According to the experimental results, too much or too little of a magnetic abrasive filling quantity can not obtain low surface roughness or suitable material-removal thickness, and the ideal state is achieved when the magnetic abrasive filling quantity is 0.15–0.2 g. When the magnetic abrasive filling quantity is 0.1 g, the material-removal thickness is higher, which indicates that the magnetic abrasive filling quantity is related to the stability and stiffness of the FMAB. When the magnetic abrasive filling quantity is too small, the stiffness of the FMAB increases and the stability decreases, which makes it easy to remove the defect layer of the inner wall of the tube, but it is also easy to scratch the surface, resulting in increased surface roughness. When the magnetic abrasive filling quantity exceeds 0.2 g, the surface roughness and material-removal thickness change greatly. This indicates that too much MAPs results in being prone to clogging in the tube, which cannot better maintain the finishing state, resulting in the incomplete removal of the defect layer. At the same time, multiple MAPs cannot participate in the MAF process, resulting in waste.

### 4.7. Surface Observation of the Inner Wall before and after Magnetic Abrasive Finishing

Figure 11 is the SEM image of the original morphology of the inner wall of the cobalt–chromium alloy cardiovascular stent tube. It can be seen from Figure 11 that the surface is relatively rough and that there are obvious pits, folds, and scratches. A cardiovascular stent with these defects may present a risk of secondary thrombosis after long-term implantation. The surface roughness *Ra* of the original inner wall is 0.485 μm measured by a 3D digital microscope. When the finishing time is 360 min, the tube rotational speed is 900 r/min, the feed speed of the magnetic pole is 150 mm/min, the abrasive size is 150 μm, and the magnetic abrasive filling amount is 0.2 g; the surface roughness *Ra* is 0.101 μm measured by 3D digital microscope. The SEM image of the inner wall of the tube is shown in Figure 12. It is obvious that the defects in the inner wall of cobalt–chromium alloy cardiovascular stent tubes are removed and the surface becomes smoother, which is of great significance to improving the biocompatibility of cardiovascular stents.

## 5. Conclusions

(1)In this study, the proposed magnetic abrasive finishing setup for the inner wall of ultra-fine and ultra-long cardiovascular stent tubes has good stability, which can solve the problem of finishing the inner wall of different materials in future research. At the same time, the CBN/metal spherical magnetic abrasive powder prepared by gas–solid two-phase double-stage atomization and rapid solidification has good finishing performance, which is suitable for the finishing of difficult-to-machine materials.(2)According to the results of single-factor experiments, the influence of various process parameters on the surface roughness *Ra* and material-removal thickness of the inner wall of the Co–Cr alloy cardiovascular stent tube processed by magnetic abrasive finishing is analyzed, which provided a reference for the optimization of process parameters in the future.(3)After 360 min of magnetic abrasive finishing, the surface roughness *Ra* of the inner wall of a cobalt–chromium alloy cardiovascular stent tube decreases from 0.485 μm to 0.101 μm. Magnetic abrasive finishing technology is used to solve the problem of poor surface quality of the inner wall of cobalt–chromium alloy cardiovascular stent tubes, which is of great significance to practical applications.

## Figures and Tables

**Figure 1 micromachines-13-01374-f001:**
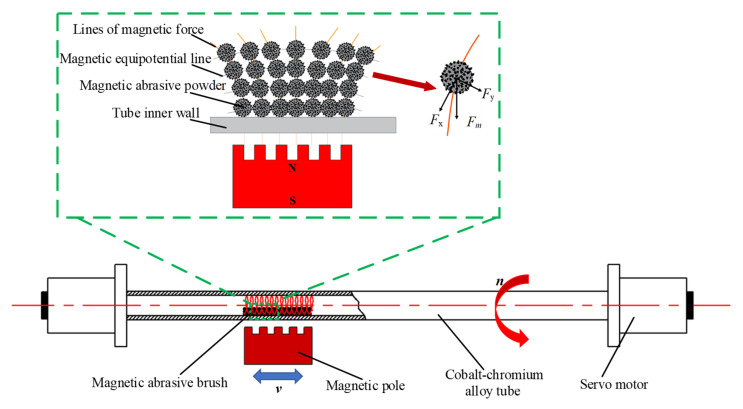
Principle of magnetic abrasive finishing.

**Figure 2 micromachines-13-01374-f002:**
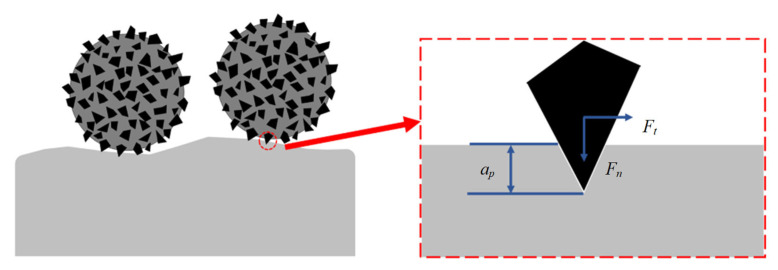
Diagram of material removal.

**Figure 3 micromachines-13-01374-f003:**
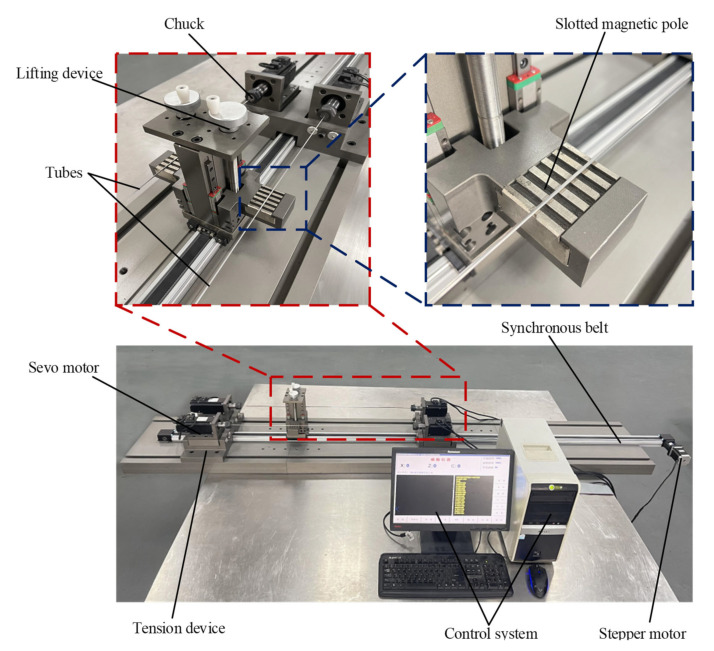
Experimental setup.

**Figure 4 micromachines-13-01374-f004:**
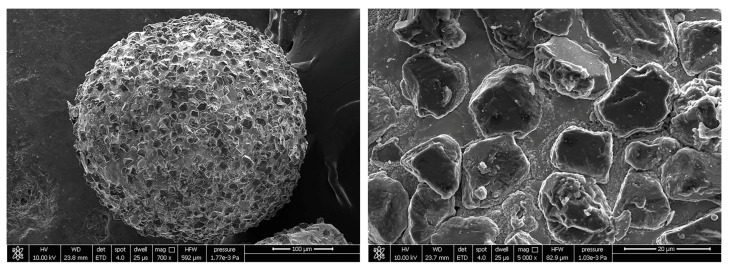
SEM image of a spherical CBN/metal MAP.

**Figure 5 micromachines-13-01374-f005:**
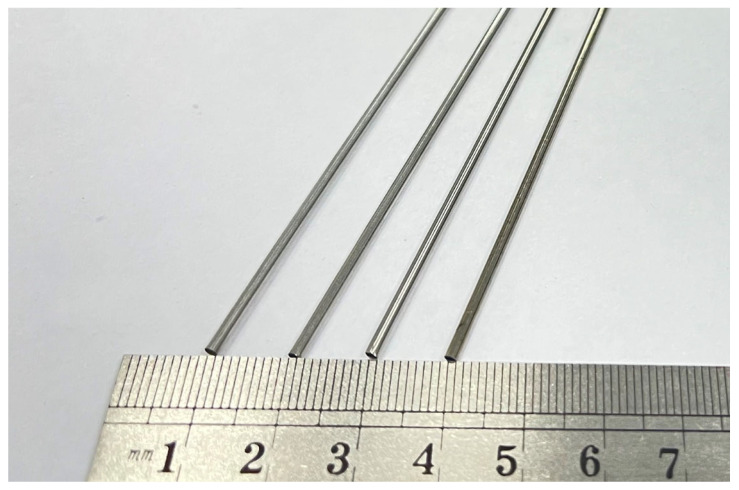
Cardiovascular stent tube of the cobalt–chromium alloy used in the experiment.

**Figure 6 micromachines-13-01374-f006:**
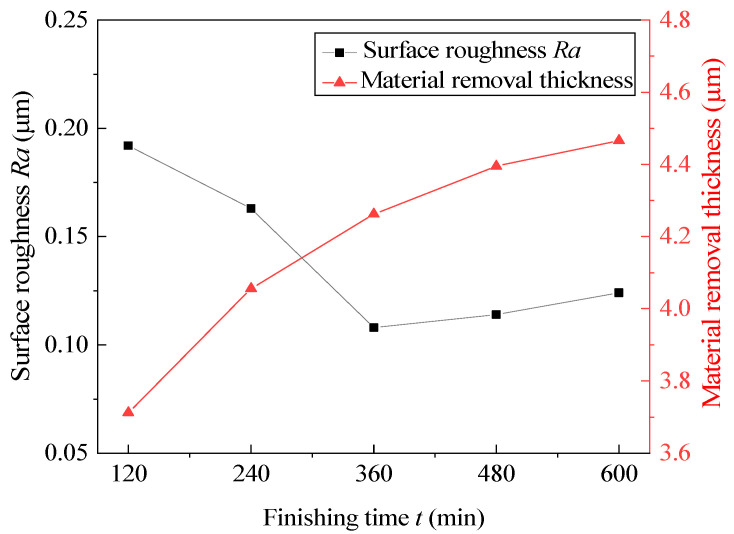
Effect of finishing time on surface roughness and material-removal thickness.

**Figure 7 micromachines-13-01374-f007:**
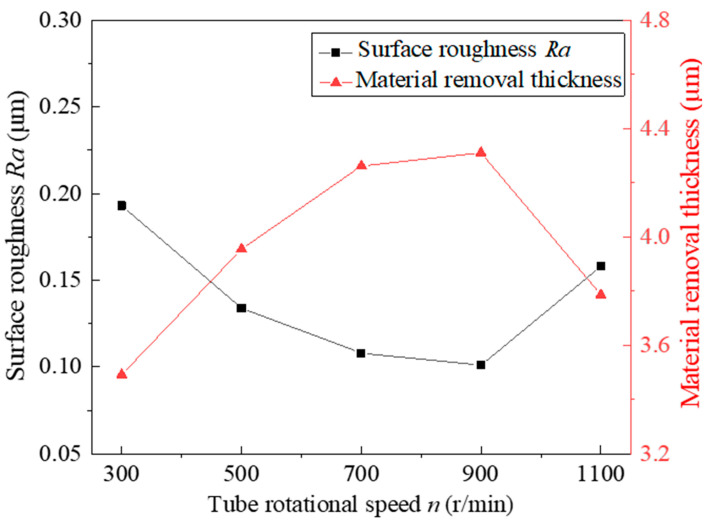
Effect of tube rotational speed on surface roughness and material-removal thickness.

**Figure 8 micromachines-13-01374-f008:**
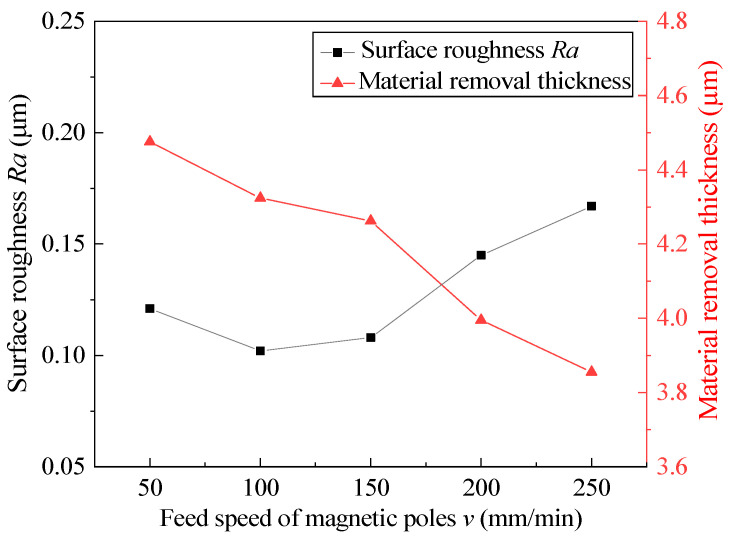
Effect of the feed speed of the magnetic pole on surface roughness and material removal thickness.

**Figure 9 micromachines-13-01374-f009:**
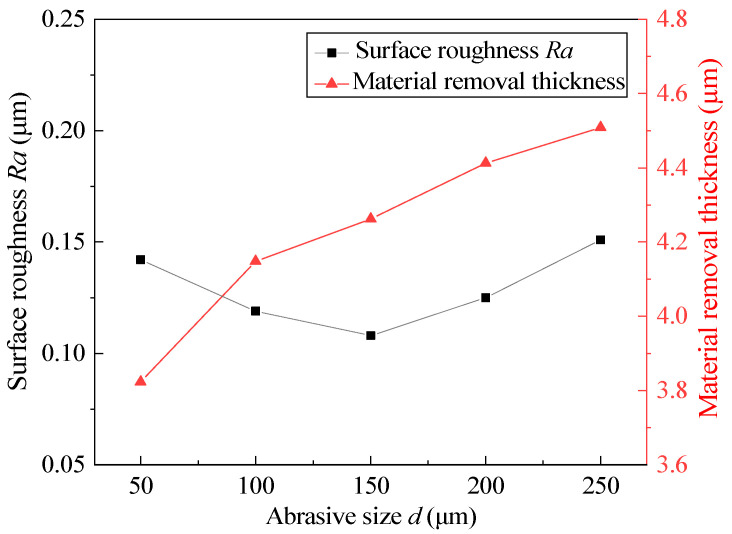
Effect of abrasive size on surface roughness and material-removal thickness.

**Figure 10 micromachines-13-01374-f010:**
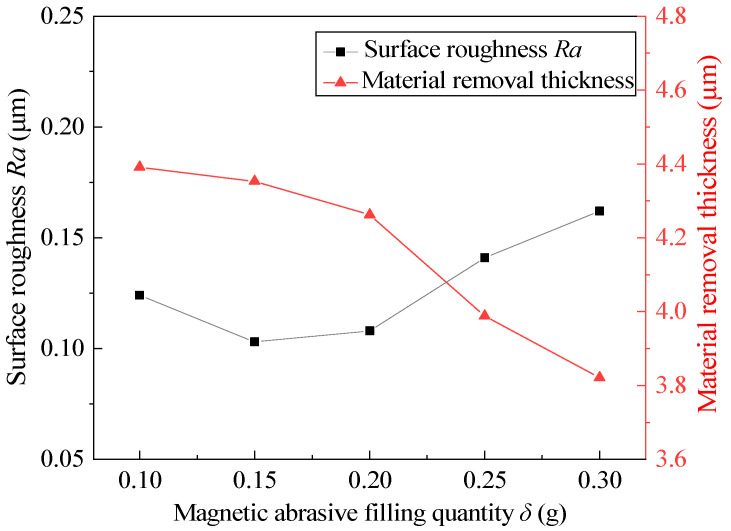
Effect of the magnetic abrasive filling quantity on surface roughness and material-removal thickness.

**Figure 11 micromachines-13-01374-f011:**
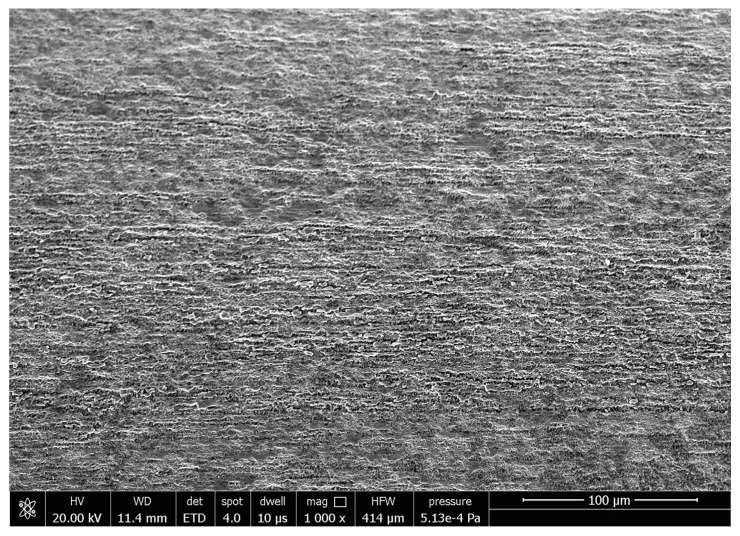
SEM image before magnetic abrasive finishing.

**Figure 12 micromachines-13-01374-f012:**
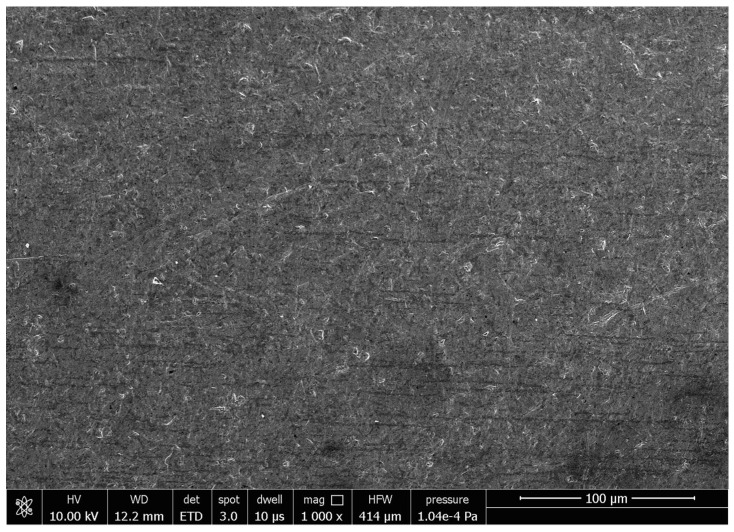
SEM image after magnetic abrasive finishing.

**Table 1 micromachines-13-01374-t001:** Composition of the Co–Cr alloy tube.

Element	Cr	W	Ni	Fe	C	Si	Mn	Co
w/%	19~21	14~16	9~11	≤3	≤0.15	≤1	≤2	Bal.

**Table 2 micromachines-13-01374-t002:** Performance indexes of the Co–Cr alloy tube.

Performance Indicators	Density (g·cm^3^)	Elastic Modulus (Gpa)	Tensile Strength (MPa)	Yield Strength (MPa)	Elongation (%)	Elastic Range (%)
Value	9.10	243	820~1200	420~600	35~55	0.16~0.32

**Table 3 micromachines-13-01374-t003:** Single-factor experimental design.

Process Parameters	Levels				
	1	2	3	4	5
Finishing time *t*/(min)	120	240	360	480	600
Tube rotational speed *n* (r/min)	300	500	700	900	1100
Feed speed of magnetic pole *v* (mm/min)	50	100	150	200	250
Magnetic abrasive filling quantity *δ*/(g)	0.1	0.15	0.2	0.25	0.3
Abrasive size *d*/(μm)	50	100	150	200	250

## Data Availability

Not applicable.

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
