# Peer review of "Study on the Micro Removal Process of Inner Surface of Cobalt Chromium Alloy Cardiovascular Stent Tubes"

_micromachines, 2022, doi:10.3390/mi13091374_

Round 1

Reviewer 1 Report

In this paper the authors studied effect of parameters like a) rotational speed, b) polishing/finishing time, c) magnetic pole feed, d) abrasive size in magnetic abrasive particles, and e) magnetic abrasive particles quantity on surface finish of inner wall of the tubes. It is an interesting work; however reviewer recommends several changes before the paper is accepted. Please look at the suggestions/questions below:

1. The abstract should justify the need for this research work. In abstract, you need to explain the literature gap, why this work is important and what is the new knowledge from your research. Therefore, I recommend rewriting the abstract to explain to readers about the knowledge gap in literature and how your work is filling the knowledge gap in literature.

2. The literature review is not sufficient. Reviewer suggests doing a detailed literature review related to magnetic abrasive finishing and discuss atleast few papers from work done by Yamaguchi et al., VK Jain et al., Shanbhag et al. etc.  in the literature section in the revised manuscript.

3. The section “Principle of processing” is not required as this knowledge can be found elsewhere like books, online articles, or state-of-art review. Author should focus more on experimental work, results, and discussion.

4. All the readers may not know what CBN is. Please include full form of CBN and also check for missing full form of other acronym in paper.

5. Did the author consider on having slotted magnetic pole on both sides of the tube? As per research conducted by Naveen K et al., a N-S configuration of magnetic tool gives far better surface finish when compared to have a magnet only on one side.  

Naveen, K., Shanbhag, V.V., Balashanmugam, N. and Vinod, P., 2017. Investigation of magnetic abrasive finishing using unbonded magnetic abrasives with double pole arrangement. International Journal of Manufacturing Technology and Management31(4), pp.314-326.

6. The use of gas-solid two-phase double-stage atomization and rapid solidification method is interesting. As many readers may not be aware about this technique, review recommends to explain how this process was used to prepare magnetic abrasive particles.

7. For each point that is shown in graph (e.g., figure 6), how many readings were taken to calculate the average?

8. Why was CBN abrasives preferred and why not diamond abrasives? As per literature, diamond abrasives gives mirror like surface finish in very short short duration.  

Naveen, K., Shanbhag, V.V., Balashanmugam, N. and Vinod, P., 2018. Ultra-precision finishing by Magnetic Abrasive Finishing process. Materials Today: Proceedings, 5(5), pp.12426-12436.

9. For how long was each test done in figure 7? Was the same sample used for every test or new sample used for every test? Similar question also for figure 8, 9 and 10.

10. Please include a discussion section and compare the work done in literature to your work.

11. Please check for grammatical mistakes throughout the paper. 

Author Response

1.The abstract should justify the need for this research work. In abstract, you need to explain the literature gap, why this work is important and what is the new knowledge from your research. Therefore, I recommend rewriting the abstract to explain to readers about the knowledge gap in literature and how your work is filling the knowledge gap in literature.

Answer:
Thanks for your valuable advice. I have revised the abstract according to my opinion and marked it as red.

2.The literature review is not sufficient. Reviewer suggests doing a detailed literature review related to magnetic abrasive finishing and discuss atleast few papers from work done by Yamaguchi et al., VK Jain et al., Shanbhag et al. etc. in the literature section in the revised manuscript.

Answer:
Thanks for your valuable advice. I have re-added the literature review in the introduction and marked red in the original text.

  1. The section “Principle of processing” is not required as this knowledge can be found elsewhere like books, online articles, or state-of-art review. Author should focus more on experimental work, results, and discussion.

Answer:
Thanks for your valuable advice. The purpose of adding the section “ processing principle ” is to make the article more complete, which can prevent readers from not understanding how to carry out magnetic abrasive finishing on the inner wall of the tube.

  1. All the readers may not know what CBN is. Please include full form of CBN and also check for missing full form of other acronym in paper.

Answer:
Thanks for your valuable advice. I introduced CBN in the introduction and labeled it red

  1. Did the author consider on having slotted magnetic pole on both sides of the tube? As per research conducted by Naveen K et al., a N-S configuration of magnetic tool gives far better surface finish when compared to have a magnet only on one side.  

Naveen, K., Shanbhag, V.V., Balashanmugam, N. and Vinod, P., 2017. Investigation of magnetic abrasive finishing using unbonded magnetic abrasives with double pole arrangement. International Journal of Manufacturing Technology and Management, 31(4), pp.314-326.

Answer:
Thanks for your valuable advice. The processing principle of this study is shown in the figure below. The magnetic pole releases enough strong magnetic induction intensity to adsorb the magnetic abrasive on the inner wall of the tube, and the tube rotates driven by the motor. The magnetic abrasive does not move with the rotation of the pipe, so that a more regular thread trajectory will be generated. If two magnetic poles are used, the magnetic induction intensity inside the pipe will be enhanced, but the magnetic abrasive is prone to chaos inside the tube, which leads to uneven machining effect. The questions raised by reviewers are valuable, and we will also reform the equipment and conduct experimental research later.

  1. The use of gas-solid two-phase double-stage atomization and rapid solidification method is interesting. As many readers may not be aware about this technique, review recommends to explain how this process was used to prepare magnetic abrasive particles.

Answer:
Thanks for your valuable advice. There have been special reports on the preparation methods of magnetic abrasives, such as “Surface roughness prediction and process parameter optimization of Ti‑6Al‑4V by magnetic abrasive finishing”. Therefore, this is not the focus of this report.

  1. For each point that is shown in graph (e.g., figure 6), how many readings were taken to calculate the average?

Answer:
Thanks for your valuable advice. Surface roughness Ra is the average of five measured positions

  1. Why was CBN abrasives preferred and why not diamond abrasives? As per literature, diamond abrasives gives mirror like surface finish in very short short duration.

Naveen, K., Shanbhag, V.V., Balashanmugam, N. and Vinod, P., 2018. Ultra-precision finishing by Magnetic Abrasive Finishing process. Materials Today: Proceedings, 5(5), pp.12426-12436.

Answer:
Thanks for your valuable advice. The reason why we choose CBN abrasive is verified by experiments. It is found that CBN abrasive is more suitable for the processing of cobalt chromium alloy materials and more economical.

  1. For how long was each test done in figure 7? Was the same sample used for every test or new sample used for every test? Similar question also for figure 8, 9 and 10.

Answer:
Thanks for your valuable advice. Each group of experiments in Figs. 7, 8, 9 and 10 was carried out for 360 min. The discussion on this aspect was omitted, which was re-discussed and marked as red in the new manuscript.

  1. Please include a discussion section and compare the work done in literature to your work.

Answer:
Thanks for your valuable advice. There are few studies on the inner wall of cardiovascular direct tube, and it is still unable to compare with other literature. We will conduct more detailed research on this topic.

  1. Please check for grammatical mistakes throughout the paper.

Answer:
Thanks for your valuable advice. I have coloured the language of the manuscript.

Reviewer 2 Report

1. About the format and language:

- The manuscript was well prepared. However, there remain some grammatical errors and typos. For example, the first sentences in the abstract “In order to remove the surface defects of cobalt-chromium allow cardiovascular stent tube and improve the surface quality of its inner wall” is not completed. The authors may find the same issues in the other sections, such as 4.1.

- Some terms were not used properly. I don’t think a “new setup” can be “invented”. I highly recommend terms such as “presented”, “proposed”, “developed” … instead of “invented”.

- The authors are recommended to change the paper title to a more proper expression. The expression “study on” is usually used in a general situation where most of the aspect of a problem is investigated. In my opinion, this work is not enough to review all the aspects or issues in this field.

2. About the quality of the work

(1) In the Introduction section, the authors introduced some publications relating to your work. For example, Heng et al [13] reduced the roughness from 0.18µm to 0.02µm which is far better than your results (0.101µm). So what is your improvement here?

(2) Please elaborate why it is very difficult to finish the inner wall of cobalt-chromium alloy cardiovascular stent tube with inner diameter of 1.6mm and outer diameter of 1.88 by traditional method? Is it either so dedicated situation or any special reason?

(3) Fig.1 express the configuration setup for the finishing method. The right section was noted as servo motor. So what is the left section? Do you use 2 servos? What for?

(4) In this work, MAP seems to be the key. In the abstract, the authors express steps to implement the work. However, the content does not reveal any information about the first step – “CBN/metal spherical magnetic abrasive powders with high performance are prepared by gas-solid two-phase double stage atomization and rapid solidification method”.

(5) the Last issue is about the experimental design. There are 5 important factors that need to investigate. In each section, the authors tried to adjust one parameter and remain others, then all the adjustment is related to one single target – the surface roughness. I think the experiment should be implemented with other experimental designs, such as Taguchi, or Principle Component Analysis (PCA) to reveal the best condition in this dataset.

Author Response

  1. About the format and language:

- The manuscript was well prepared. However, there remain some grammatical errors and typos. For example, the first sentences in the abstract “In order to remove the surface defects of cobalt-chromium allow cardiovascular stent tube and improve the surface quality of its inner wall” is not completed. The authors may find the same issues in the other sections, such as 4.1.

- Some terms were not used properly. I don’t think a “new setup” can be “invented”. I highly recommend terms such as “presented”, “proposed”, “developed” … instead of “invented”.

Answer:
Thanks for your valuable advice. I have coloured the language of the manuscript.

- The authors are recommended to change the paper title to a more proper expression. The expression “study on” is usually used in a general situation where most of the aspect of a problem is investigated. In my opinion, this work is not enough to review all the aspects or issues in this field.

Answer:
Thanks for your valuable advice. I 've changed my title to ' Study on micro removal process of inner surface of cobalt chromium alloy cardiovascular stent tubes'.

  1. About the quality of the work

(1) In the Introduction section, the authors introduced some publications relating to your work. For example, Heng et al [13] reduced the roughness from 0.18µm to 0.02µm which is far better than your results (0.101µm). So what is your improvement here?

Answer:
Thanks for your valuable advice.At present, there are few reports on the use of magnetic particle finishing technology to process the inner wall of ultra-thin and long pipes. In the introduction, in order to prove the efficiency of the magnetic particle finishing technology, some recent reports on this technology are cited. These reports are different from our processing materials and methods. At the same time, these reports all study the finishing of the outer surface of the material, and we study the finishing of the inner wall. Therefore, we just to illustrate the efficiency of magnetic particle finishing technology, which is the reason why we apply this technology to the inner wall finishing of cobalt chromium alloy cardiovascular stent tube.

(2) Please elaborate why it is very difficult to finish the inner wall of cobalt-chromium alloy cardiovascular stent tube with inner diameter of 1.6mm and outer diameter of 1.88 by traditional method? Is it either so dedicated situation or any special reason?

Answer:
Thanks for your valuable advice. Because of the small inner diameter and long length of cobalt chromium alloy cardiovascular stent tube, the traditional finishing tool cannot penetrate the tube. At the same time, the outer wall can not be damaged in the finishing process, which leads to the traditional processing method can not remove the defect layer of the inner wall. I ' ve modified it in the introduction and marked it red.

(3) Fig.1 express the configuration setup for the finishing method. The right section was noted as servo motor. So what is the left section? Do you use 2 servos? What for?

Answer:
Thanks for your valuable advice. The device on the left is also a servo motor. The length of cobalt chromium alloy cardiovascular stent used in the experiment is 500 mm, and the inner diameter of the stent is very small. If there is only one servo motor, the coaxiality of the tube in the machining process cannot be guaranteed, which leads to uneven machining effect.

(4) In this work, MAP seems to be the key. In the abstract, the authors express steps to implement the work. However, the content does not reveal any information about the first step – “CBN/metal spherical magnetic abrasive powders with high performance are prepared by gas-solid two-phase double stage atomization and rapid solidification method”.

Answer:
Thanks for your valuable advice. There have been special reports on the preparation methods of magnetic abrasives, such as “Surface roughness prediction and process parameter optimization of Ti‑6Al‑4V by magnetic abrasive finishing”. Therefore, this is not the focus of this report. I have revised the abstract according to opinion and marked it as red.

(5) the Last issue is about the experimental design. There are 5 important factors that need to investigate. In each section, the authors tried to adjust one parameter and remain others, then all the adjustment is related to one single target – the surface roughness. I think the experiment should be implemented with other experimental designs, such as Taguchi, or Principle Component Analysis (PCA) to reveal the best condition in this dataset.

Answer:
Thanks for your valuable advice. In this study, we designed a single factor experiment to explore the effects of various process parameters on surface roughness and material removal thickness, which provides experimental parameters for future design optimization experiments. According to the results of this report, our research on parameter optimization has been presented in other reports and is being reviewed.

Round 2

Reviewer 1 Report

The author has not accepted most of the suggested changes. For example, 1) in the revised literature only 1 additional literature was added. It needs to be extensive. 2) Justification for use of CBN abrasives over diamond abrasives is not provided in the paper. 3) Explanation on use of gas-solid two-phase double-stage atomizaion and rapid solidification method to prepare magnetic abrasives is not given. This is very important. Therefore, reviewer recommends to revisit the paper and make changes recommended in round-1 by the reviewer so that it benefits the new readers and also it improves quality of manuscript. 

Author Response

1) in the revised literature only 1 additional literature was added. It needs to be extensive.

Answer:
Thanks for your valuable advice. In the new manuscript, two new references are cited. The modified place is marked red in the original text.

2) Justification for use of CBN abrasives over diamond abrasives is not provided in the paper.

Answer:
Thanks for your valuable advice. Cubic boron nitride (CBN) has the advantages of high hardness, high compressive strength and high thermal conductivity, which makes the machined surface not easy to crack and burn. In addition, the chemical inertness of CBN is strong, and it is not easy to react with iron group elements. Therefore, for the finishing of cobalt-chromium alloy, the selection of CBN hard abrasive particles can obtain better surface quality. In the new manuscript, the above is added. The modified place is marked red in the original text.

3) Explanation on use of gas-solid two-phase double-stage atomizaion and rapid solidification method to prepare magnetic abrasives is not given.

Answer:
Thanks for your valuable advice. The brief preparation process of this method is added, and the literature first proposed by this method is cited. The modified place is marked red in the original text.

Reviewer 2 Report

The manuscript has been revised and improved. I recommend to publish the current manuscript. 

Author Response

Thank the reviewers for their valuable suggestions.

Round 3

Reviewer 1 Report

The author has accepted all the changes suggested by the reviewer. It is an interesting paper which will benefit the scientific community and industries to understand more about achieving very fine surface finish in inner wall of the tubes.